# Photodynamic and Cold Atmospheric Plasma Combination Therapy Using Polymeric Nanoparticles for the Synergistic Treatment of Cervical Cancer

**DOI:** 10.3390/ijms22031172

**Published:** 2021-01-25

**Authors:** Ji-Hui Ha, Young-Jin Kim

**Affiliations:** Department of Biomedical Engineering, Daegu Catholic University, Gyeongsan 38430, Korea; bme13160878@gmail.com

**Keywords:** photodynamic therapy, cold atmospheric plasma, combination therapy, polymeric nanoparticles, cervical cancer

## Abstract

Integrating multi-modal therapies into one platform could show great promise in overcoming the drawbacks of conventional single-modal therapy and achieving improved therapeutic efficacy in cancer. In this study, we prepared pheophorbide a (Pheo a)/targeting ligand (epitope analog of oncoprotein E7, EAE7)-conjugated poly(γ-glutamic acid) (γ-PGA)/poly(lactide-co-glycolide)-block-poly(ethylene glycol) methyl ether (MPEG-PLGA)/hyaluronic acid (PPHE) polymeric nanoparticles via self-assembly and encapsulation method for the photodynamic therapy (PDT)/cold atmospheric plasma (CAP) combinatory treatment of human papillomavirus (HPV)-positive cervical cancer, thereby enhancing the therapeutic efficacy. The synthesized PPHE polymeric nanoparticles exhibited a quasi-spherical shape with an average diameter of 80.5 ± 17.6 nm in an aqueous solution. The results from the in vitro PDT efficacy assays demonstrated that PPHE has a superior PDT activity on CaSki cells due to the enhanced targeting ability. In addition, the PDT/CAP combinatory treatment more effectively inhibited the growth of cervical cancer cells by causing elevated intracellular reactive oxygen species generation and apoptotic cell death. Moreover, the three-dimensional cell culture model clearly confirmed the synergistic therapeutic efficacy of the PDT and the CAP combination therapy using PPHE on CaSki cells. Overall, these results indicate that the PDT/CAP combinatory treatment using PPHE is a highly effective new therapeutic modality for cervical cancer.

## 1. Introduction

Cervical cancer is the fourth common type of cancer and the fourth leading cause of cancer-related mortality in women worldwide [1]. The high-risk human papillomavirus (HPV) infection, which is found in almost all cancer tissues from patients with cervical cancer, is the major cause of cervical cancer progression and development [2,3]. Among the various genotypes of HPV, HPV 16 and 18 are the leading causes of cervical cancer. Cancer has an incidence rate of approximately 70% [4]. The oncogenic transformation by HPV is mediated by the viral HPV E6 and HPV E7 oncogenes. Oncoprotein E7 binds to the retinoblastoma protein (pRb), resulting in the degradation and function inactivation of the pRb tumor suppressor gene [5,6,7]. Therefore, oncoprotein E7 is a crucial factor in HPV-related carcinogenesis and promote abnormal cell proliferation, leading to cancer occurrence [1,8,9]. Oncoprotein E7 is particularly always retained and expressed in HPV-positive cancer cells.

Despite their various adverse side effects, radical hysterectomy and concurrent chemoradiotherapy are still favored modalities for curing cervical cancer, which can decrease the patient’s quality of life [3,10]. These treatment modalities cannot preserve fertility. In addition, chemoradiotherapy has been shown to induce deoxyribonucleic acid (DNA) damage in normal cells, resulting in cell destruction. Thus, alternative cancer treatment methods for reducing side effects and preventing cancer recurrence and metastasis have been extensively investigated [10,11]. One treatment modality in cervical cancer therapy is photodynamic therapy (PDT), non-invasive and conservative therapeutic option for fertility preservation. The main elements of PDT are photosensitizers, specific-wavelength light, and molecular oxygen [12]. The photosensitizers are irradiated by a specific-wavelength light after having been administered and absorbed by the cancer cells. The excited photosensitizers then interact with the molecular oxygen present in the tissue to produce reactive oxygen species (ROS), such as free radicals, singlet oxygen (^1^O_2_), and triplet oxygen. The PDT cytotoxic properties are essentially due to the produced ROS, which can induce the oxidation of a large range of biomolecules in cells, including proteins, DNA, and lipids, resulting in cancer cell death [10,13].

The major advantages of the PDT treatment are the noninvasiveness and the minimization of damage to adjacent healthy normal tissues [14]. However, most photosensitizers present high hydrophobicity and form aggregates in aqueous media and body fluids [15]. These hydrophobic properties can affect their photophysical, chemical, and biological effects, thereby limiting their clinical applications. In addition, this PDT system generally suffers from unsatisfactory tumor specificity and the premature leakage of phototherapeutic drugs during blood circulation due to the limited tumor-targeting capability, which causes phototoxic effects on normal cells [12,16]. Therefore, delivery systems must be used to overcome the drawback related to the high hydrophobicity and the lack of the target specificity of photosensitizers [17].

To solve the abovementioned problems, polymeric nanoparticles have been extensively studied to enhance the treatment effect and reduce the detrimental side effects of cancer therapy [12,16,18]. However, these nanoparticles still also exhibit low treatment effectiveness because of their limited blood circulation and cellular internalization [19]. Thus hydrophilic polymers (e.g., polyethylene glycol (PEG) and polysaccharides) have been adopted to form the shell of polymeric nanoparticles and improve blood circulation [12,20]. Some other strategies have also been employed to enhance the intracellular uptake of polymeric nanoparticles by incorporating cancer-targeting ligand and reducing the nanoparticle size [21,22].

Despite efforts extended toward solving the relevant problems, the adverse side effects of PDT still remain. Therefore, the combination of PDT with other therapeutic modalities, such as chemotherapy, photothermal therapy, and plasma therapy, has been explored to overcome its drawbacks, including the unsatisfactory efficacy of the PDT treatment originating from the insufficient ROS generation caused by a hypoxic tumor microenvironment [16,23,24]. This combination can hold great potential to achieve an outstanding antitumor efficacy. Among them, plasma therapy using cold atmospheric plasma (CAP) has emerged as a selective cancer treatment with a high affinity for triggering death in cancer cells, while exhibiting no harmful effect on normal cells [23,25,26]. CAP is a partially ionized gas, including diverse charged molecules, ultraviolet radiation, reactive species, and neutral molecules. The most probable CAP therapeutic effects are primarily linked to high ROS concentrations and reactive nitrogen species, which are responsible for the antiproliferative cell mechanisms and cell death in cancer cells through the oxidative damage of the cell membrane and the DNA [25,26].

The viral oncoprotein E7 is highly conserved and only constitutively expressed in HPV-infected or HPV-transformed cells, but not in normal cells [27]. Oncoprotein E7 can bind to the human epidermal growth factor receptor 3 (HER3) overexpressed in HPV-positive cancer cells [28,29]. Hyaluronic acid (HA) has been widely used as an active targeting ligand due to its specific interaction with the receptor CD44 overexpressed in various malignant tumors [24]. Many studies have demonstrated that HA-based nanoparticles can undergo rapid internalization into various cancer cells through both enhanced permeability and retention effect and CD44-mediated endocytosis [16,24].

In the present study, we developed a new drug delivery system of biodegradable polymer-based polymeric nanoparticles via a layer-by-layer (LBL) self-assembly method for the HER3/CD44 dual-targeted PDT/CAP combinatory treatment of HPV-positive cervical cancer. The polymeric nanoparticles are composed of pheophorbide a (Pheo a)-conjugated poly(γ-glutamic acid) (γ-PGA) (γ-PGA-Pheo a), amphiphilic copolymer poly(lactide-co-glycolide)-block-poly(ethylene glycol) methyl ether (MPEG-PLGA), and targeting ligand (epitope analog of oncoprotein E7, EAE7, PEG2-86TLGIVCPI93)-decorated HA (HA-EAE7). The complexes of γ-PGA-Pheo a and MPEG-PLGA were self-assembled into polymeric nanoparticles as the inner cores of nanoparticles. Subsequently, these cores were successfully encapsulated with HA-EAE7 shell, resulting in Pheo a/EAE7-conjugated γ-PGA/MPEG-PLGA/HA (PPHE) polymeric nanoparticles (Figure 1). A stable nanoparticle structure was maintained via amide bonding, hydrogen bonding, and hydrophobic interactions. The physicochemical properties and in vitro cytotoxicity of the PPHE polymeric nanoparticles were systematically characterized to prove the availability of the polymeric nanoparticles for cancer therapy. In addition, PPHE polymeric nanoparticles were intravenously injected into tumor-bearing mice to evaluate in vivo imaging and ex vivo biodistribution. The enhanced therapeutic effects of using the combination of PDT with CAP therapy on cervical cancer were also intensively investigated. Accordingly, a more exact therapeutic efficacy of the combinative method was investigated using a three-dimensional (3D) cancer cell culture model.

## 2. Results and Discussion

### 2.1. Preparation and Properties of the PPHE Polymeric Nanoparticles

The hydrophilicity and the targeting ability of polymeric nanoparticles are major factors for enhancing the therapeutic efficacy of drugs and reducing the detrimental side effects. Figure 1 schematically illustrates the preparation process of the versatile PPHE polymeric nanoparticles and their application in the PDT/CAP combination therapy for cervical cancer. Aminated Pheo a-conjugated γ-PGA and EAE7-decorated HA were synthesized through carbodiimide reaction to prepare the HER3/CD44 dual-targeting PPHE polymeric nanoparticles. The final PPHE polymeric nanoparticles containing γ-PGA-Pheo a, MPEG-PLGA, and HA-EAE7 were prepared via the LBL self-assembly method. The structural analysis of γ-PGA-Pheo a and HA-EAE7 was performed by ^1^H nuclear magnetic resonance (NMR) spectroscopy, which exhibited the expected characteristic peaks (Appendix A). The Pheo a signals were observed at 9.58, 9.32, 9.11, 8.34, 6.45, 6.18, 4.57, 4.40, 3.68, 3.53, 3.39, 3.31, 2.31, and 2.03 ppm. The peaks were observed at 3.75 and 1.65 ppm, which were associated with γ-PGA. The peaks corresponding to EAE7 were exhibited at 4.47, 4.21, 3.94, 3.81, 3.50, 3.23, 2.25, 1.68, 1.24, and 0.91 ppm, while that assigned to HA was seen at 2.04 ppm. The degree of conjugation, which is defined as the number of Pheo a or EAE7 per repeating unit of γ-PGA or HA, was estimated as 0.06 for Pheo a and 0.19 for EAE7.

MPEG-PLGA is an amphiphilic block copolymer that can self-assemble into a core–shell type of polymeric nanoparticle with a controlled size at the nanometer level [30,31]. These polymeric nanoparticles are suitable for drug delivery because of their excellent biocompatibility, appropriate stability, and high drug-loading content. Therefore, the PPHE polymeric nanoparticles were readily prepared via self-assembly and encapsulation in aqueous solutions and exhibited a quasi-spherical shape with an average diameter of 71.5 ± 6.7 nm (Figure 2a). The size of the PPHE polymeric nanoparticles was confirmed through the DLS method. The nanoparticles were almost monodispersed with narrow size distribution and a diameter of 80.5 ± 17.6 nm (Figure 2b).

The enhanced hydrophilicity of the PPHE polymeric nanoparticles was validated using the solubility test in dimethyl sulfoxide (DMSO) and Dulbecco’s phosphate-buffered saline (DPBS). Pheo a had good solubility in DMSO, but was not soluble in the aqueous media because of its high hydrophobicity (Appendix A). However, PPHE exhibited an improved solubility in DPBS, and the characteristic absorption peaks of the UV–visible spectra were observed at 403, 508, 610, and 672 nm.

The physiological pH in the bloodstream is 7.4 and the pH value of intracellular lysosome is 4.5 [32]. The ideal controlled drug delivery system requires the ability to suppress drug release during circulation in blood vessels but fast release the loaded drug in the targeted cells. The PPHE polymeric nanoparticles exhibited an initial burst release of Pheo a, followed by a prolonged gradual increase of release (Appendix A). Pheo a was released faster from the PPHE polymeric nanoparticles at pH 4.5 than at pH 7.4 in the DPBS solution because of pH-dependent cleavage of amide bonds.

### 2.2. In Vitro Phototoxicity of the PPHE Polymeric Nanoparticles

The in vitro phototoxicity of free Pheo a and PPHE polymeric nanoparticles on the CaSki and HCT116 cells after treatment with or without laser irradiation was investigated using the MTT assay to evaluate the PDT efficacy of the PPHE polymeric nanoparticles. Free Pheo a displayed considerably low phototoxicity on the cancer cells at all tested concentrations (Figure 3a,b). Even at a high Pheo a dose of 8 μg/mL, the cells remained approximately 85 and 78% viable after treatment with free Pheo a and laser irradiation, respectively, in the CaSki and HCT116 cells. This result indicates that free Pheo a could not effectively kill the cancer cells. In addition, the PPHE polymeric nanoparticles exhibited low phototoxicity on the HCT116 cells after the laser irradiation (Figure 3a). However, these PPHE polymeric nanoparticles exhibited a remarkable cell-killing ability on the CaSki cells in a dose-dependent manner upon light irradiation (Figure 3b). Approximately 75% of the targeted CaSki cells were killed by 8 μg/mL of PPHE, showing an enhanced PPHE phototoxicity on the CaSki cells.

The cell viability of the CaSki and HCT116 cells were visualized by a simultaneous double staining method with the LIVE/DEAD Viability/Cytotoxicity Assay Kit. Figure 3c,d depict that the fluorescence variation in the CaSki and HCT116 cells displayed a similar variation tendency with the MTT assay results. The HCT116 cells showed almost green fluorescence after treatment with PPHE and laser irradiation, implying that the PPHE polymeric nanoparticles had a negligible effect on the cells (Figure 3c). However, the CaSki cells were efficiently killed upon treatment with PPHE and laser irradiation (Figure 3d).

As mentioned earlier, oncoprotein E6 and E7 can bind to HER3 overexpressed in HPV-positive cancer cells [28,29]. Therefore, a competitive cytotoxicity assay was performed using CaSki cells to verify the improved phototoxicity of the PPHE polymeric nanoparticles upon immobilization with the targeting ligand EAE7. The cancer cells were preincubated with or without excess free EAE7 for 1 h and then treated with PPHE and laser irradiation. The CaSki cell viability was determined using the MTT assay, which displayed that the addition of the excess free EAE7 reduced the PPHE phototoxicity compared with the case without preincubation with free EAE7 (Appendix A). Consequently, the targeting ability of EAE7 toward cervical cancer cells caused the enhanced phototoxicity of the PPHE polymeric nanoparticles.

The enhanced phototoxicity of PPHE was proven by assaying the nanoparticle intracellular uptake. The nanoparticle internalization in cancer cells is the foremost step in PDT treatment, but their efficiency is often hindered by off-target effects [24]. Therefore, a dual-targeting strategy was adopted herein in the PPHE polymeric nanoparticles to enhance their intracellular uptake ability. HA and EAE7 were employed as the targeting moieties to interact with CD44 and HER3 overexpressed in the CaSki cells. The intracellular uptake of free Pheo a and PPHE into the CaSki cells was studied using a flow cytometer and a confocal laser scanning microscope, resulting in a higher intracellular uptake of PPHE compared with free Pheo a in the CaSki cells (Appendix A).

### 2.3. In Vitro PDT and CAP Combination Therapy

Even though the PDT is an important noninvasive therapeutic modality in cancer therapy, it also exhibits adverse side effects, such as phototoxic damage to healthy normal tissues and an unsatisfactory therapeutic efficacy by insufficient ROS generation [12,14,16]. We tried using the combination of PDT with CAP therapy to enhance the therapeutic efficacy on cervical cancer and solve the abovementioned problems. As expected, the CaSki cell viability was significantly reduced by the combination of PDT with CAP treatment at all tested concentrations of free Pheo a and PPHE, indicating that the PDT/CAP combinatory treatment effectively inhibited the cell proliferation in a dose-dependent manner (Figure 4a). The cell viability after the PDT treatment with free Pheo a decreased from 80 to 60% by subsequent CAP treatment. Lower cell viability (53 and 40%) was also achieved, even at low PPHE concentrations of 1 and 2 μg/mL. In other words, the PDT/CAP combinatory modality could more effectively kill cancer cells.

The main therapeutic effects of the PDT and CAP treatment on cancer cells are primarily related to ROS production. ROS can cause significant cytotoxicity upon reaching an intracellular threshold concentration, which leads to cell death via apoptosis or necrosis [33]. Therefore, sufficient ROS production is very important for the effective treatment of cancer cells. The intracellular ROS generation in the CaSki cells was quantitatively measured herein by detecting the green fluorescent DCF using a flow cytometer. Figure 4b depicts that the PDT treatment with free Pheo a or PPHE produced a small amount of ROS. However, the PDT/CAP combinatory treatment resulted in a remarkably increased DCF fluorescence intensity due to the elevated ROS production, which led to a considerably enhanced cytotoxicity on the CaSki cells.

The ROS-mediated PDT and CAP treatment can induce cell apoptosis with negligible adverse effects to the adjacent cells [10,13,23]. This apoptotic cell death can be measured by the externalization of phosphatidylserine and DNA through binding annexin V and PI. Annexin V may recognize apoptotic cells, but PI can discriminate the late apoptotic and necrotic cells [24]. Accordingly, annexin V/PI staining and flow cytometry analysis was performed to confirm the cell death mechanism induced by the PDT/CAP combinatory treatment. Figure 5a displays that the combination of the PDT with CAP treatment induced higher apoptotic cell death compared to the single PDT. The cells treated with the PDT/CAP combinatory treatment using PPHE exhibited the highest apoptotic cell population, indicating that the combined PDT/CAP treatment could kill cancer cells in an apoptosis-accelerating manner. The rapid ROS generation in cancer cells causes mitochondrial damage, the release of proapoptotic proteins into cytosol, and DNA fragmentation, resulting in apoptotic cell death [34]. This is advantageous because apoptotic cell death occurs without the drastic effect of inflammation observed in necrosis.

The CaSki cells were also stained with annexin V and PI after the PDT/CAP combinatory treatment and observed using a fluorescence microscope (Figure 5b). The fluorescence in the cancer cells treated with the PDT/CAP combinatory treatment using PPHE exhibited a green color caused by the binding of annexin V to the phosphatidylserine translocated to the outer membrane layer in the apoptotic cells [18,35]. In addition, a red fluorescence indicating that the PI stained the nucleus in the apoptotic cells was observed. The H-33258 stained cancer cells after the PDT/CAP combinatory treatment using free Pheo a or PPHE were observed to investigate the nuclear morphologic characteristics. Contrary to the control or only the PDT-treated cancer cells, many chromatin fragmentations, multinucleation, and dot-like chromatin condensation were observed in the cancer cells treated with the PDT/CAP combinatory treatment using the PPHE polymeric nanoparticles (Figure 5b), which are all apoptotic features [10,20]. These results clearly prove the cell apoptosis induction by the combined PDT/CAP treatment.

The expression of the apoptosis-related proteins in cancer cells was measured by a western blot analysis to further confirm the cell death mechanism. The protein expression of the cleaved PARP and the cleaved caspase-3 markedly increased after the PDT/CAP combinatory treatment using the PPHE polymeric nanoparticles (Figure 6). The results confirmed that the PPHE polymeric nanoparticles are highly effective PDT agents, and the PDT/CAP combinatory treatment is a novel modality for cervical cancer therapy.

### 2.4. 3D Cancer Cell Culture Model

The two-dimensional (2D) cell culture model of cancer cells is a leading methodology for evaluating the anticancer efficacy of drugs. However, this in vitro monolayer assay can neither form stratified cancer cell colonies nor exactly evaluate the drug efficacy on the invasion and metastasis of cancer cells [36]. A conventional 2D cell culture model cannot completely recapitulate the complex pathological 3D microenvironment in tumor tissues due to the lack of biological, chemical, and structural cues, leading to inaccurate or even wrong predictive data. Therefore, the 3D culture of cancer cells has long been advocated as a better model of the malignant phenotype.

We first fabricated the PLGA/γ-PGA/poly(propylene glycol)-poly(ethylene glycol)-poly(propylene glycol) triblock copolymer (Pluronic 17R4) porous scaffolds with a bimodal pore structure via the thermally-induced phase separation (TIPS) method to prepare a 3D cancer cell culture model to investigate the exact efficacy of the PDT/CAP combinatory treatment on the CaSki cells. Figure 7a depicts that the resulting scaffold exhibited a three-dimensionally interconnected open pore structure. The surface and the cross-section comprised a bimodal pore structure. The cell morphology and the interaction between the cells and the scaffolds were observed by SEM after culturing for 15 days. The CaSki cells adhered and spread in the scaffold surface. The cells then migrated into the macropores and covered the pores (Figure 7b). These results strongly suggest that the 3D culture of the CaSki cells can be successfully achieved using porous scaffolds.

The cytotoxicity of the PDT/CAP combinatory treatment on the 3D cell culture model of the CaSki cells was evaluated by the MTT assay. The result exhibited a similar pattern with the results from the 2D cell culture model. Figure 8a shows that only the PDT-treated CaSki cells with free Pheo a or PPHE (4 μg/mL Pheo a) exhibited higher cell viability (94.6% for free Pheo a and 60.0% for PPHE) in the 3D cell culture model compared with that (84.9% for free Pheo a and 46.7% for PPHE) in the 2D cell culture model, indicating that the increased resistance of cancer cells to anticancer drugs in the 3D cell culture system when compared with the 2D cell culture system. The structural architecture in a 3D tumor tissue model regulated the differentiated cell functions through the changes in cell shapes and increased the cell–cell and cell–matrix interactions [37]. These changes in the cancer cell function extensively affected the response of a tumor tissue model on external drugs, resulting in cancer cells exhibiting increased resistance to anticancer drugs. However, the PDT/CAP combinatory treatment in the 2D and 3D cell culture models displayed almost the same therapeutic efficacy. The cancer cell viability decreased to below 30% after the PDT/CAP combinatory treatment using PPHE, indicating that the combined PDT and CAP therapy exhibited an obvious therapeutic activity on the CaSki cells.

CaSki cells were incubated and stained with a LIVE/DEAD Viability/Cytotoxicity Assay Kit to visually evaluate the synergistic therapeutic efficacy of the PDT/CAP combinatory treatment. They were then examined using a confocal laser scanning microscope (Figure 8b). We observed the cancer cells stained with EthD-1 (red) and enormously decreased the cell viability after the PDT/CAP combinatory treatment using the PPHE polymeric nanoparticles. The results show that this combinatory modality on cervical cancer can probably display a higher therapeutic efficacy, even at in vivo models, because the in vitro 3D cell culture can bridge the gap between an extremely simplified 2D cell culture and complex animal models. In addition, the transport limitations generate an in vivo-like response to external drugs by restricting accessibility and including cellular heterogeneity in 3D cell culture models.

### 2.5. Biodistribution and Imaging Assays

The ideal therapeutic nanoparticles should accumulate at the tumor site at a specific time and then be excreted from the body. Therefore, the long blood circulation lifetime of the nanoparticles is usually a prerequisite for efficient cancer treatment. The fluorescence signal produced by Pheo a was recorded to visually display their tumor-targeting capability at preset times of 2, 6, 8, and 24 h after the intravenous injection of free Pheo a or PPHE polymeric nanoparticles. Figure 9a shows that the PPHE-injected mouse exhibited a strong red or yellow fluorescence at the early time points due to the superior blood circulation of PPHE. A higher fluorescence signal in the tumor site was observed at 2 h post-injection. In addition, the high fluorescence signal level was sustained until 24 h after the injection, which was approximately four times higher than the fluorescence signal of free Pheo a (Figure 9b). These results confirmed the effective intratumoral accumulation of the PPHE polymeric nanoparticles. In contrast, free Pheo a revealed a weak fluorescence signal caused by the non-specific accumulation in the tumor site and the rapid excretion from the body at all time points. A comparison of the two sets of results suggested that the PPHE polymeric nanoparticles had an enhanced tumor-targeting capability.

At 24 h post-injection, the major organs and the tumors in the PPHE treatment group were excised to assay the nanoparticle biodistribution. Figure 9c,d illustrate that the PPHE polymeric nanoparticles were mainly accumulated in the tumor tissue, but free Pheo a exhibited a poor ability to accumulate in the tumor tissue and major organs. As revealed in the ex vivo fluorescence images of the PPHE polymeric nanoparticles, stronger and weaker fluorescence was observed in the tumor tissue and the heart at 24 h after the PPHE injection. In addition, the fluorescence signal in the other organs, except for the lung, became weaker in the order of liver, kidney, and spleen. The higher fluorescence signal in the lung was probably caused by the PPHE accumulation within the capillaries of the lung by the PPHE aggregation behavior [38]. This selective PPHE accumulation in the tumor site can allow its use for subsequent cervical cancer treatment by PDT.

## 3. Materials and Methods

### 3.1. Materials

Pheo a was purchased from Frontier Scientific (Logan, UT, USA). MPEG-PLGA (MPEG 2 kDa, PLGA 3 kDa, LA/GA = 50/50) was purchased from Nanosoft Polymers (Winston-Salem, NC, USA). γ-PGA (20–50 kDa) was obtained from Wako Pure Chemical Industries (Osaka, Japan). PLGA (LA/GA = 50/50) was purchased from Corbion (Amsterdam, The Netherlands). These materials were used without further purification. EAE7 for cervical cancer-targeting was obtained from Anygen (Gwangju, Korea). Ethylene diamine, HA sodium salt from Streptococcus equi (8–15 kDa), Pluronic 17R4 (2.7 kDa), N-(3-dimethylaminopropyl)-N′-ethylcarbodiimide hydrochloride (EDC), N-hydroxysuccinimide (NHS), 3-(4,5-dimethyl-2-thiazolyl)-2,5-diphenyl-2H-tetrazolium bromide (MTT), 3-amino-7-dimethylamino-2-methylphenazine hydrochloride (neutral red), bisBenzimide H 33258 (Hoechst 33258, H-33258), and DMSO were obtained from Sigma–Aldrich (St. Louis, MO, USA).

The human cervical cancer cell lines (CaSki) and the human colon cancer cell lines (HCT116) were obtained from the American Type Culture Collection (Manassas, VA, USA). The LIVE/DEAD Viability/Cytotoxicity Assay Kit, 2′,7′-dichlorodihydrofluorescein diacetate (DCF-DA), and SlowFade Gold antifade mountant were purchased from Molecular Probes (Eugene, OR, USA). The Actin Cytoskeleton and Focal Adhesion Staining Kit was obtained from Merck Millipore (Burlington, MA, USA). The FITC Annexin V Apoptosis Detection Kit was obtained from BD Biosciences (Franklin Lakes, NJ, USA). The other reagents and solvents were commercially obtained and used as received.

### 3.2. Preparation of the PPHE Polymeric Nanoparticles

The PPHE polymeric nanoparticles were synthesized as follows: Pheo a was first aminated to conjugate onto γ-PGA through the amide bond formation. Pheo a (88.9 mg) was dissolved in 20 mL DMSO, then added with EDC (47.9 mg) and NHS (28.8 mg) to activate its carboxylic groups. Ethylene diamine (30.1 mg) was then added to the mixture, and the reaction was proceeded to synthesize the aminated Pheo a at 25 °C for 24 h. Subsequently, the aminated Pheo a conjugation onto γ-PGA was conducted by the amide bond formation. γ-PGA (96.8 mg), EDC (239.6 mg), and NHS (143.8 mg) were dissolved in 25 mL DMSO for 6 h before adding a solution of aminated Pheo a (105.8 mg) dissolved in 20 mL DMSO. The reactants were stirred at 25 °C for 24 h to synthesize γ-PGA-Pheo a. The products were dialyzed using a tubular membrane in a mixture of DMSO and distilled water (DW) for 48 h to remove unreacted agents. This was followed by lyophilization in vacuo. Similar to the γ-PGA-Pheo a synthesis, the targeting ligand EAE7-decorated HA (HA-EAE7) was synthesized by amide linkage formation. HA (121.3 mg), EDC (74.7 mg), and NHS (44.9 mg) were dissolved in 20 mL DW for 6 h. After which, EAE7 (144 mg) dissolved in 15 mL acetonitrile/DW at a 1:3 ratio was added to the mixture and reacted at 25 °C for 24 h to produce HA-EAE7. The products were dialyzed in DW for 48 h, followed by lyophilization in vacuo.

A two-step LBL self-assembly method was used to fabricate the PPHE polymeric nanoparticles. First, γ-PGA-Pheo a (103.2 mg) was dissolved in 5 mL DMSO. MPEG-PLGA (29.8 mg) was then dissolved in 6 mL acetonitrile/DW at a 1:2 ratio. These solutions were added dropwise to 100 mL DW and vigorously stirred at 25 °C for 12 h to prepare the self-assembled γ-PGA-Pheo a/MPEG-PLGA (PP) nanoparticles. Subsequently, the as-prepared HA-EAE7 (27.3 mg), dissolved in 5 mL, DW was added dropwise to the PP dispersed solution and vigorously stirred for 6 h to encapsulate the PP cores of the polymeric nanoparticles as a shell. The final products were dialyzed in DW for 12 h. The resultant PPHE polymeric nanoparticles were then isolated by centrifugation, followed by lyophilization in vacuo. The amount of Pheo a conjugated to γ-PGA was determined by measuring the absorbance at 405 nm using an ultraviolet–visible (UV–visible) spectrometer. The amount of the targeting ligand EAE7 decorated onto HA was also measured using a fluorescamine assay [39].

### 3.3. Characterization of the PPHE Polymeric Nanoparticles

The γ-PGA-Pheo a and HA-EAE7 structures were analyzed by ^1^H NMR spectroscopy (ADVANCE III 400, Bruker BioSpin, Billerica, MA, USA). The morphology of the resultant PPHE polymeric nanoparticles was observed by transmission electron microscopy (TEM, H-7600, Hitachi, Tokyo, Japan) after sputter-coating the samples with platinum. The average diameter of the PPHE polymeric nanoparticles was determined by analyzing the TEM images with Image-Pro Plus (Media Cybernetics Inc., Rockville, MD, USA). In addition, the particle size distribution of the PPHE polymeric nanoparticles was determined by the dynamic light scattering (DLS) technique using a Zetasizer Nano ZS (Malvern Instruments, Malvern, UK). The UV–visible spectra were recorded on a Hitachi U-2900 spectrometer (Tokyo, Japan), while the fluorescence emission spectra were measured using a Perkin-Elmer LS55 spectrofluorophotometer (Waltham, MA, USA) at room temperature.

The in vitro release studies of Pheo a from the PPHE polymeric nanoparticles were performed using a dialysis method in a thermostatic shaking incubator (NB-205, N-BIOTEK, Bucheon, Korea). A weighed amount (10 mg) of the PPHE polymeric nanoparticles was dispersed in 10 mL DPBS and then transferred into a dialysis membrane (molecular weight cut-off 2 kDa). The dialysis membrane was immersed into 100 mL DPBS (pH 4.5 or 7.4) and placed in a shaking incubator (200 rpm, 37 °C). The supernatant was collected from the DPBS solution at preset time points. The cumulative release amount of Pheo a from the PPHE polymeric nanoparticles was determined by measuring the absorption of the samples at 403 nm using a UV–visible spectrometer. The percentage of released Pheo a was then calculated based on the initial weight of Pheo a conjugated in the PPHE polymeric nanoparticles.

### 3.4. Cell Culture

CaSki and HCT116 cells were cultured in RPMI-1640 supplemented with 10% fetal bovine serum (FBS) and 0.5% penicillin-streptomycin. The cells were maintained at 37 °C in a humidified incubator with 5% CO_2_. They were detached using 0.25% trypsin-ethylenediaminetetraacetic acid (EDTA) for passage.

### 3.5. In Vitro Intracellular Uptake Tests

The in vitro intracellular uptake of the free Pheo a and the PPHE was first quantified using a flow cytometer (FACSCalibur™, BD Biosciences, Franklin Lakes, NJ, USA). The CaSki cells were seeded into six-well plates at densities of 2 × 10^5^ cells per well and cultured at 37 °C for 24 h. Subsequently, these cells were treated with free Pheo a or PPHE polymeric nanoparticles (6 μg/mL Pheo a) for 2 h before washing twice with DPBS. Subsequently, the cells were trypsinized and re-suspended in DPBS after centrifugation (1500 rpm, 3 min). The collected cells were analyzed using a flow cytometer. The in vitro cellular uptake and distribution of drugs in the CaSki cells were visualized using an inverted LSM 700 confocal laser scanning microscope (Carl Zeiss, Oberkochen, Germany) after staining with H-33258 and tetramethylrhodamine conjugated phalloidin (phalloidin-TRITC) for 30 min in the dark.

### 3.6. In Vitro Phototoxicity Assays

The MTT assay was employed to evaluate the phototoxicity of Pheo a and the PPHE polymeric nanoparticles in the CaSki and HCT116 cells. The cells (1 × 10^4^ cells per well) were seeded into 96-well plates and cultured at 37 °C for 24 h. The medium was replaced by 0.2 mL RPMI-1640 containing predetermined concentrations of free Pheo a or PPHE polymeric nanoparticles (0–8 μg/mL Pheo a). Next, the cells were incubated for another 2 h and rinsed twice with DPBS. The cells were then irradiated with a 671 nm laser (42 mW/cm^2^, 1 min) after adding fresh culture medium to each well. The final irradiated cell viability was determined by the MTT assay after culturing for another 24 h. The culture medium was mixed 20 μL of the MTT solution (5 mg/mL in DPBS), and the cells were incubated for another 3 h. Next, the remaining medium was removed, and 0.3 mL DMSO was added to solubilize the precipitated formazan crystals. Finally, 0.1 mL triplicates from each resulting sample were transferred to 96-well plates, and the optical density at 570 nm was determined using a microplate reader (OPSYS-MR, Dynex Technology Inc., Chantilly, VA, USA).

The CaSki and HCT116 cell viability after the laser irradiation was qualitatively analyzed using the LIVE/DEAD Viability/Cytotoxicity Assay Kit according to the manufacturer’s instructions. In this assay, the calcein AM stains live cells green, while ethidium homodimer-1 (EthD-1) stains dead cells red [40]. The CaSki and HCT116 cells (5 × 10^4^ cells per well) were seeded into 12-well plates and cultured at 37 °C for 24 h. The medium was then replaced by 2 mL of RPMI-1640 containing 6 μg/mL of free Pheo a or PPHE polymeric nanoparticles. Next, the cells were incubated for another 2 h and rinsed twice with DPBS. Thereafter, the cells were irradiated with a 671 nm laser (42 mW/cm^2^, 1 min) after adding fresh culture medium to each well and cultured for another 24 h. The cells were stained for 30 min at room temperature with 1 μM calcein AM and 2 μM EthD-1, followed by another 24 h of incubation. Live and dead cells were observed using a fluorescence microscope (Eclipse TS100, FITC-G2A filters, Nikon, Tokyo, Japan).

A competitive phototoxicity test was conducted to prove the cervical cancer cell targeting ability of the PPHE polymeric nanoparticles. CaSki cells were seeded into 96-well plates at a density of 1 × 10^4^ cells per well and incubated at 37 °C for 22 h. The cells were then preincubated with or without free EAE7 (10 μg/mL) for 2 h. Subsequently, these cells were treated with the PPHE polymeric nanoparticles (0–8 μg/mL Pheo a) for another 2 h and rinsed twice with DPBS. The cells were then irradiated with a 671 nm laser (42 mW/cm^2^, 1 min) after adding fresh culture medium to each well and cultured further for 24 h. The final viability of irradiated cells was determined by MTT assay.

### 3.7. In Vitro PDT and CAP Combination Therapy

The cytotoxicity of the combined PDT and CAP treatment on the CaSki cells was evaluated by the MTT assay. CaSki cells (1 × 10^4^ cells per well) were seeded into 96-well plates and cultured at 37 °C for 24 h. The medium was replaced by 0.2 mL RPMI-1640 containing predetermined concentrations of free Pheo a or PPHE polymeric nanoparticles (0–6 μg/mL Pheo a). Next, the cells were incubated for another 2 h and rinsed twice with DPBS. They were then treated with a 671 nm laser (42 mW/cm^2^, 1 min) and CAP (argon flow rate = 3 L/min; discharge voltage = 15 kV; frequency = 34 kHz; spot size of the plasma jet = 5 mm; exposure time = 10 s) after adding fresh culture medium to each well. The condition of the CAP treatment, excluding exposure time, was fixed and used in the subsequent experiment. The distance between the nozzle tip and the cells was fixed at 1.5 cm when actuating. The final irradiated cell viability was determined by the MTT assay after culturing for another 24 h.

### 3.8. Evaluation of the Intracellular ROS Generation

The intracellular ROS generation was investigated using a flow cytometer. CaSki cells (1 × 10^5^ cells per well) were seeded into 12-well plates and cultured at 37 °C for 48 h. The cells were then treated with free Pheo a or PPHE polymeric nanoparticles (4 μg/mL Pheo a) for another 2 h. Next, they were washed twice with Hank’s balanced salt solution (HBSS) and treated with a 671 nm laser (42 mW/cm^2^, 1 min) and CAP (exposure time = 10 s), followed by treatment for 30 min at 37 °C with 0.5 mL DCF-DA (20 μM) in HBSS. DCF-DA is a fluorogenic marker for ROS, permeates live cells, and is deacetylated by intracellular esterases to produce fluorescent DCF. Finally, the ROS concentration trapped by DCF-DA was quantitatively measured using a flow cytometer (excitation, 488 nm; emission, 530 nm). Untreated CaSki cells (no drug, no laser, and no CAP) were detected as control. In addition, the cells treated with the drug before treatment with a laser and CAP were used as comparison groups.

### 3.9. Apoptosis and Necrosis Analysis

The apoptosis and the necrosis of CaSki cells treated with the PDT/CAP combination therapy were first analyzed using a flow cytometer. CaSki cells were seeded onto six-well plates at a density of 1 × 10^5^ cells per well and incubated for 48 h at 37 °C. These cells were treated with free Pheo a or PPHE polymeric nanoparticles (4 μg/mL Pheo a) for another 2 h and washed twice with DPBS. Following this, the cells were treated with a 671 nm laser (42 mW/cm^2^, 1 min) and CAP (exposure time = 10 s) after adding a fresh culture medium to each well. After post-incubation for 18 h, the cells were trypsinized, washed with DPBS, and stained with annexin V-FITC/propidium iodide (PI). An apoptosis and necrosis analysis was performed using a flow cytometer, and the data were analyzed using Cellquest Software (BD Biosciences, Franklin Lakes, NJ, USA). Untreated CaSki cells (no drug, no laser, and no CAP) were also detected as control.

The morphological changes of the CaSki cells were observed through fluorescence microscopy. The cells (5 × 10^4^ cells per well) were seeded into 24-well plates and incubated at 37 °C for 24 h. They were treated with free Pheo a or PPHE polymeric nanoparticles (4 μg/mL Pheo a) for another 2 h and washed twice with DPBS. Following this, the cells were treated with a 671 nm laser (42 mW/cm^2^, 1 min) and CAP (exposure time = 10 s) after adding a fresh culture medium to each well. Next, they were fixed in 4% paraformaldehyde solution for 15 min and stained with annexin V-FITC/PI and H-33258 for 30 min at room temperature in the dark. Lastly, the cells were observed through fluorescence microscopy after washing and air drying.

### 3.10. Western Blot

CaSki cells (1 × 10^6^ cells/well) were seeded onto six-well plates and cultured at 37 °C for 48 h. The cells were then treated with free Pheo a or PPHE polymeric nanoparticles (4 μg/mL Pheo a) for another 2 h, washed twice with DPBS, and treated with a 671 nm laser (42 mW/cm^2^, 1 min) and CAP (exposure time = 10 s) after adding fresh culture medium to each well. After 24 h incubation, the cells were centrifuged, rinsed twice with DPBS, and subsequently lysed in a PRO-PREP™ Protein Extraction Solution (iNtRon Biotechnology, Sungnam, Korea) containing a 1× protease and phosphatase inhibitor cocktail (Roche, Indianapolis, IN, USA). The protein concentrations were determined using a BCA Protein Assay Kit (Thermo Fisher Scientific, Waltham, MA, USA). The lysates (10–30 μg protein) were isolated by 10% sodium dodecyl sulfate–polyacrylamide gel electrophoresis (SDS–PAGE). Moreover, the resolved proteins were transferred to polyvinylidene fluoride membranes (Roche, Basel, Switzerland) blocked in Tris-buffered saline with Tween 20 and 3% skim milk for 1 h at room temperature. The blotting membranes were incubated with a primary antibody (i.e., cleaved PARP antibody or cleaved caspase-3 rabbit monoclonal antibody (Cell Signaling Technology, Danvers, MA, USA)) for 12 h. After further incubation with the secondary horseradish peroxidase-conjugated antibody at room temperature for 1 h, the immunoreactive bands were visualized by chemiluminescence detection using the WestGlow™ FEMTO Chemiluminescent Substrate (BIOMAX, Seoul, Korea). A β-actin antibody was also examined as a control to confirm the equal protein loading. The data were analyzed using the Davinch-Chemi CAS-400SM Western Imaging System and Total Lab software (Davinch-K, Seoul, Korea).

### 3.11. 3D Cancer Cell Culture Model

A 3D cancer cell culture model was prepared to prove the potential of the combined PDT and CAP treatment for application in cancer therapy. Porous scaffolds with a 3D bimodal pore structure were first fabricated via a TIPS method according to our previous report [41]. A mixture of PLGA, γ-PGA, and Pluronic 17R4 was dissolved in DMSO at 80 °C with 26 *w/v*% concentration. The weight ratio of PLGA, γ-PGA, and Pluronic 17R4 was 58:19:23. After injecting the newly obtained solution into a polytetrafluoroethylene mold, the solution was cooled to 40 °C and maintained at that temperature for 24 h for the TIPS process. The scaffolds were then cross-linked with hexamethylene diisocyanate and repeatedly washed with DW before vacuum drying. The dimensions of the final scaffolds were fixed at 15 mm diameter and 2 mm thickness.

The cytotoxicity of the combined PDT and CAP treatment on the CaSki cells cultured in the 3D system was evaluated by the MTT assay. Before the cell seeding, the scaffolds were sterilized in an autoclave at 120 °C for 20 min, placed into a 24-well tissue culture plate, and fixed with a glass ring (inner diameter = 11 mm). The CaSki cells were placed onto the sterilized scaffolds in a culture medium containing 10% FBS at densities of 1 × 10^5^ cells per well and incubated at 37.0 °C for 15 days to prepare the 3D cancer cell model. These cells were treated with free Pheo a or PPHE polymeric nanoparticles (4 μg/mL Pheo a) for another 4 h, after which the cells were washed twice with DPBS. Following this, the cells were treated with a 671 nm laser (42 mW/cm^2^, 5 min) and a CAP (exposure time = 50 s) after adding a fresh culture medium to each well. After being cultured for another 24 h, the final viability of the irradiated cells was determined by the MTT assay. The culture medium was mixed 0.1 mL of the MTT solution (5 mg/mL in DPBS), and the cells were incubated for another 3 h. Next, the remaining medium was removed, and 1 mL DMSO was added to solubilize the precipitated formazan crystals. Finally, 0.2 mL triplicates from each resulting sample were transferred to 96-well plates, and the optical density at 570 nm was determined using a microplate reader.

The CaSki cell viability after using the combined PDT and CAP treatment was visualized using the LIVE/DEAD Viability/Cytotoxicity Assay Kit. The CaSki cells (1 × 10^5^ cells per well) were seeded onto the scaffolds and cultured at 37 °C for 15 days. The medium was replaced by 1 mL of RPMI-1640 containing 4 μg/mL of free Pheo a or PPHE polymeric nanoparticles. Subsequently, the cells were incubated for another 4 h and rinsed twice with DPBS, after which the cells were irradiated with a 671 nm laser (42 mW/cm^2^, 5 min) and CAP (exposure time = 50 s) after adding fresh culture medium to each well. After being cultured for 24 h, the cells were stained with 1 μM of calcein AM and 2 μM of EthD-1, followed by another 24 h of incubation. Live and dead cells were observed using an inverted LSM 700 confocal laser scanning microscope (Carl Zeiss, Oberkochen, Germany).

### 3.12. Biodistribution and Imaging Assays

Five-week-old female BALB/c nude mice were purchased from OrientBio (Seongnam, Korea). All animal experiments were performed in accordance with the Institutional Animal Care and Use Committee guidelines of Daegu-Gyeongbuk Medical Innovation Foundation. To prepare the tumor model, 2 × 10^6^ CaSki cells in 0.1 mL DPBS were subcutaneously injected into the right flank of the mice. The subcutaneously formed tumors grew to a sufficiently large volume of 120–140 mm^3^ within 2 weeks. The tumor-bearing mice were intravenously administered with free Pheo a or PPHE at the Pheo a concentration of 5 mg/Kg via the tail vein (*n* = 6 per group). At a preset time, the mice were anesthetized and directly imaged using IVIS SpectrumCT (PerkinElmer, Waltham, MA, USA). For the in vivo fluorescence imaging, the mice were imaged at excitation and emission wavelengths of 605 and 660 nm, respectively. The mice were sacrificed at 24 h post-injection. The tumor, liver, lung, spleen, kidney, and heart were then exfoliated and imaged for the ex vivo organ biodistribution analysis using IVIS SpectrumCT.

### 3.13. Statistical Analysis

All measurements were performed at least thrice. The data are presented as the mean value ± standard deviation (SD). Two group parameters were analyzed using one-way analysis of variance, followed by Tukey’s test with SigmaPlot 13.0 (Systat Software Inc., San Jose, CA, USA). Significant results were considered as those where * *p* < 0.05.

## 4. Conclusions

The incorporation of cancer-targeting ligands and dual combinatory therapeutic approaches have emerged as a highly efficient attempt to reduce the side effects and increase the therapeutic efficacy of photosensitizers to cancer cells. We successfully developed the PDT/CAP combination therapy to enhance the therapeutic efficacy on cervical cancer using the PPHE polymeric nanoparticles, in which cancer-targeting ligands (i.e., HA and EAE7) are introduced via the LBL self-assembly method for the HER3/CD44 dual targeting of cervical cancer cells. The PPHE polymeric nanoparticles were effectively accumulated in the CaSki cells through active targeting and the passive targeting mechanism of the nanoparticle delivery system. The PDT/CAP combinatory treatment more effectively killed the cervical cancer cells by causing elevated ROS generation and induced substantial apoptosis in the cancer cells. In particular, the 3D cell culture model clearly exhibited the obvious therapeutic activity of the combined PDT and CAP therapy using PPHE on the CaSki cells. These results indicate that the PDT/CAP combinatory modality using PPHE may contribute to the development of a new therapeutic modality for cervical cancer.

## Figures and Tables

**Figure 1 ijms-22-01172-f001:**
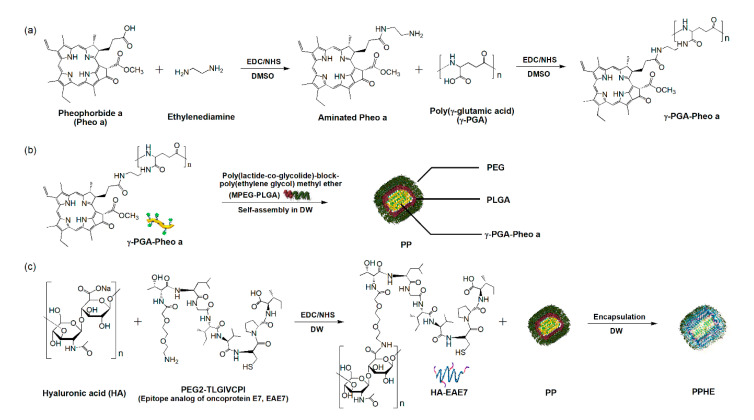
Preparation of the Pheo a/EAE7-conjugated γ-PGA/MPEG-PLGA/HA (PPHE) polymeric nanoparticles. Schematic diagram of the –synthetic process of the (**a**) Pheo a-conjugated γ-PGA (γ-PGA-Pheo a) prepared through the carbodiimide reaction to form amide linkage, (**b**) self-assembled γ-PGA-Pheo a/MPEG-PLGA (PP) nanoparticles, and (**c**) targeting ligand EAE7-decorated HA (HA-EAE7)-containing PPHE polymeric nanoparticles prepared via the amide bone formation and layer-by-layer (LBL) self-assembly method.

**Figure 2 ijms-22-01172-f002:**
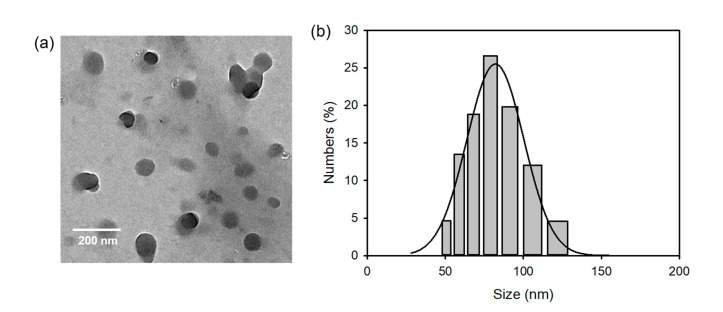
(**a**) Transmission electron microscopy (TEM) micrographs and (**b**) particle size distribution of the PPHE polymeric nanoparticles.

**Figure 3 ijms-22-01172-f003:**
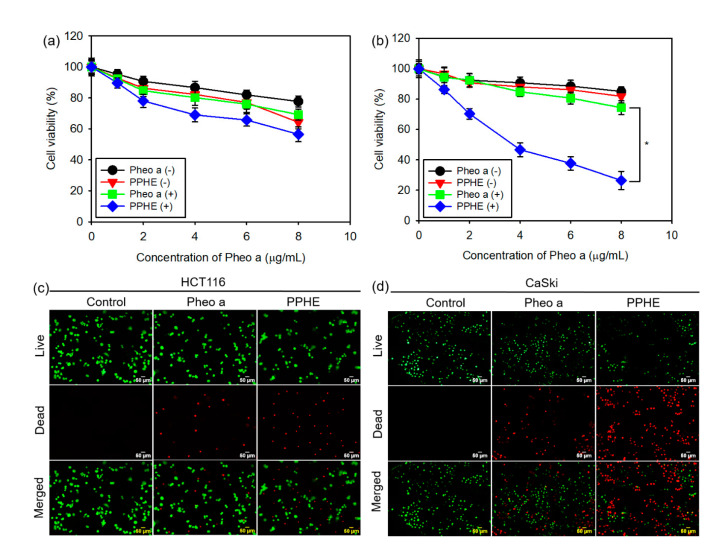
In vitro photodynamic therapy (PDT) efficacy of the free Pheo a and PPHE polymeric nanoparticles. Phototoxicity of different concentrations of the free Pheo a and PPHE polymeric nanoparticles on the (**a**) HCT116 and (**b**) CaSki cells before laser irradiation (Pheo a (−) and PPHE (−)) and after laser irradiation (Pheo a (+) and PPHE (+)) using a 671 nm laser (42 mW/cm^2^, 1 min) (*n* = 6). * *p* < 0.05 for comparison between two treatment groups. Fluorescence microscopy images of the (**c**) HCT116 and (**d**) CaSki cells after treatment with free Pheo a and PPHE polymeric nanoparticles (6 μg/mL Pheo a) and laser irradiation using a 671 nm laser (42 mW/cm^2^, 1 min). Live and dead cells were stained with calcein-AM (green) and EthD-1 (red), respectively.

**Figure 4 ijms-22-01172-f004:**
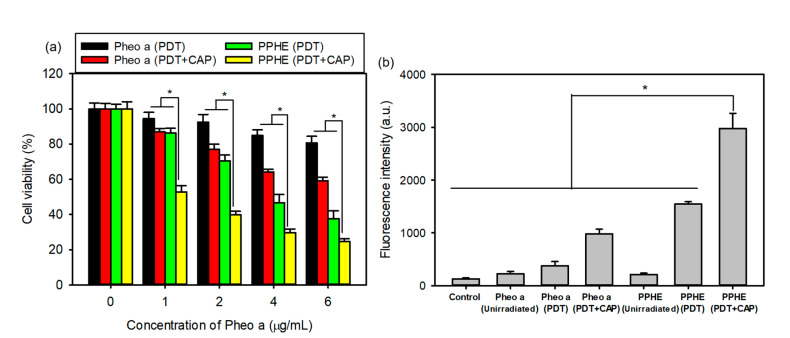
(**a**) In vitro cytotoxicity of the PDT and cold atmospheric plasma (CAP) combination therapy on the CaSki cells evaluated by the MTT assay (*n* = 6). (**b**) DCF fluorescence intensity measured using a flow cytometer for determining the intracellular reactive oxygen species (ROS) level in the CaSki cells treated with free Pheo a and PPHE polymeric nanoparticles after laser irradiation and CAP treatment (*n* = 4). * *p* < 0.05 for comparison between two treatment groups.

**Figure 5 ijms-22-01172-f005:**
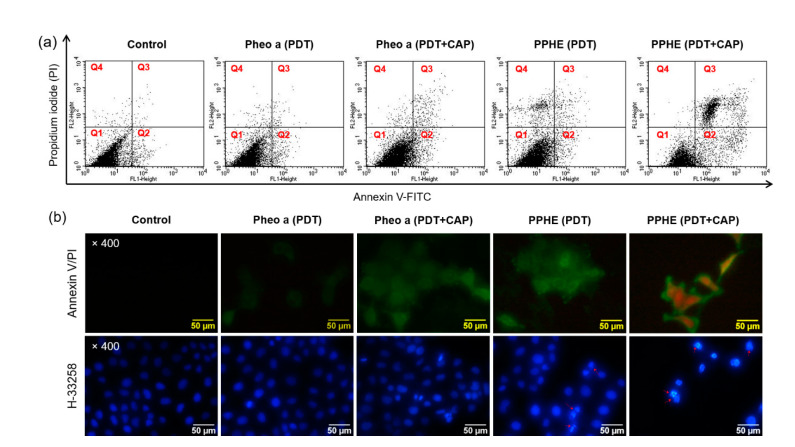
(**a**) Cell death mechanism of the PDT and CAP combination therapy. The CaSki cells were treated with free Pheo a and PPHE polymeric nanoparticles (4 μg/mL Pheo a) for 18 h after irradiation with a 671 nm laser (42 mW/cm^2^, 1 min) and CAP (exposure time = 10 s). The cells were stained with annexin V and PI and analyzed on a flow cytometer. The upper-left (Q1), upper-right (Q2), lower-left (Q3), and lower-right (Q4) quadrants in each panel indicate the population of necrosis, late apoptosis, alive, and early apoptosis, respectively. (**b**) Morphology of the CaSki cells stained with annexin V/PI and H-33258 after treatment with free Pheo a and PPHE polymeric nanoparticles (4 μg/mL Pheo a), followed by irradiation using a 671 nm laser (42 mW/cm^2^, 1 min) and CAP (exposure time = 10 s).

**Figure 6 ijms-22-01172-f006:**
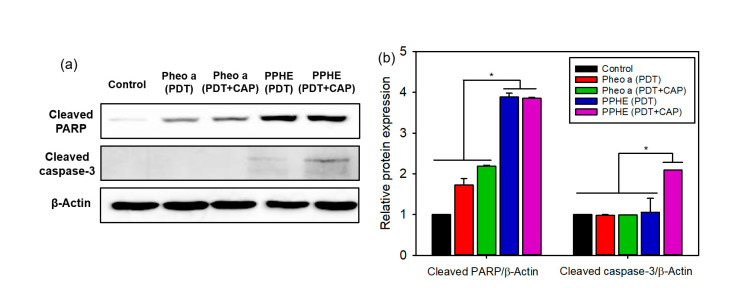
(**a**) Representative western blots of the apoptosis-related proteins and (**b**) the quantified relative expression of the apoptosis-related proteins by densitometric analysis (*n* = 6). * *p* < 0.05 for comparison between two treatment groups. β-Actin was used as the loading control.

**Figure 7 ijms-22-01172-f007:**
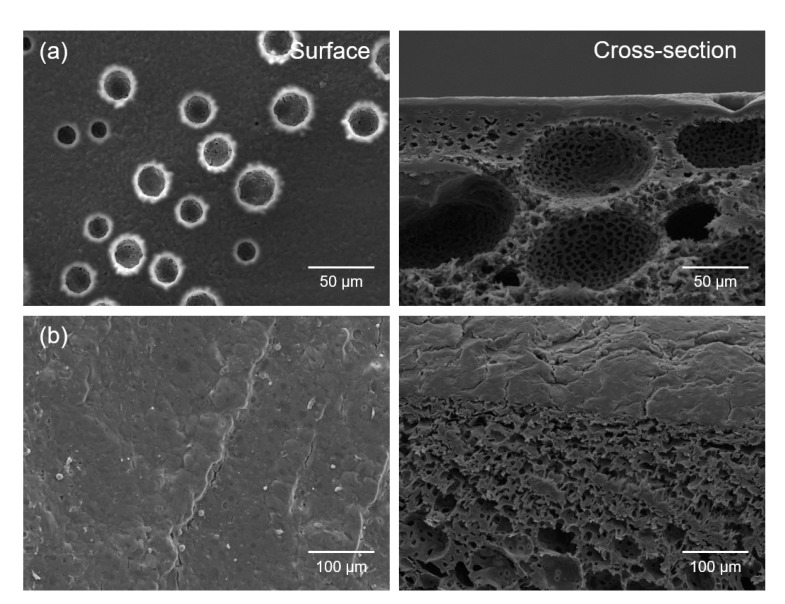
SEM micrographs of the surfaces and cross-sections of (**a**) the PLGA/γ-PGA/Pluronic 17R4 porous scaffolds with 3D bimodal pore structure fabricated via the thermally-induced phase separation (TIPS) method and (**b**) the CaSki cells grown on porous scaffolds for 15 days.

**Figure 8 ijms-22-01172-f008:**
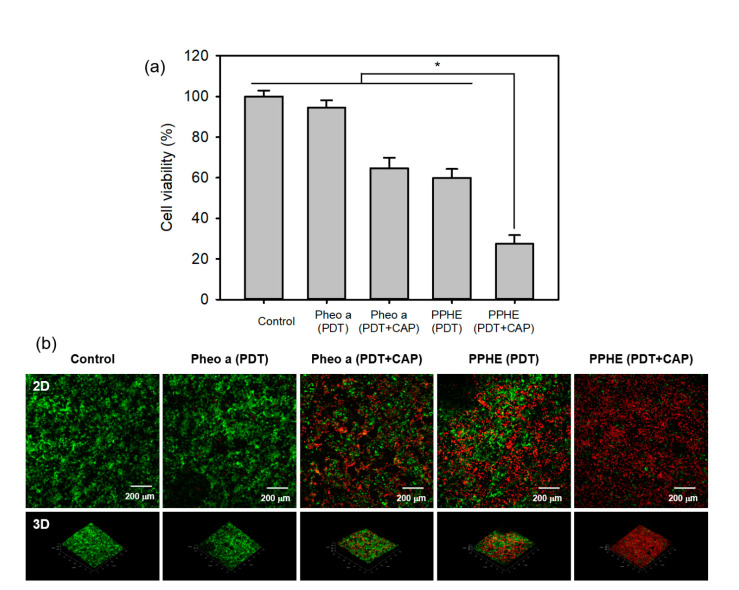
(**a**) In vitro cytotoxicity of the PDT and CAP combination therapy on the 3D cell culture model of the CaSki cells evaluated by the MTT assay (*n* = 4). * *p* < 0.05 for comparison between two treatment groups. (**b**) Fluorescence microscopy images of the 3D cell culture model of the CaSki cells after treatment with free Pheo a and PPHE polymeric nanoparticles (4 μg/mL Pheo a), followed by irradiation using a 671 nm laser (42 mW/cm^2^, 5 min) and CAP (exposure time = 50 s). Live and dead cells were stained with calcein-AM (green) and EthD-1 (red), respectively.

**Figure 9 ijms-22-01172-f009:**
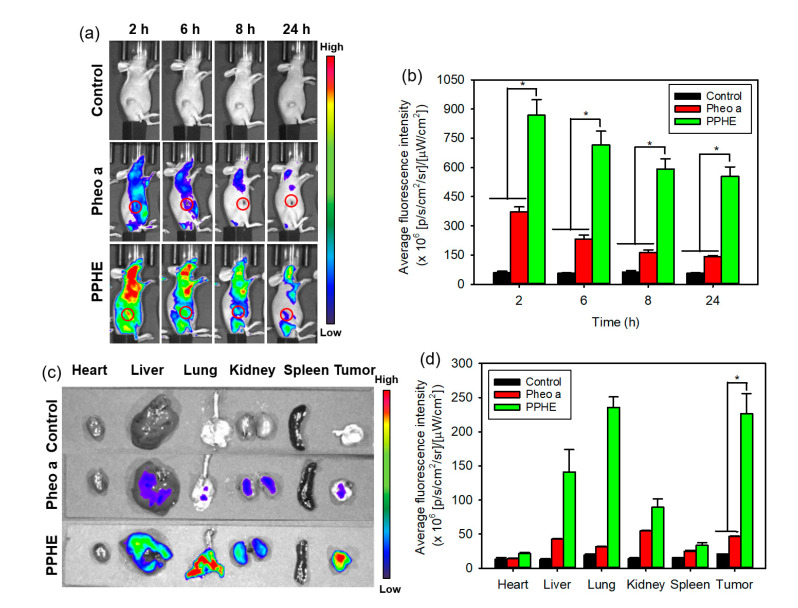
In vivo biodistribution of the free Ce6 and PPC nanoparticles. (**a**) In vivo time-dependent whole-body imaging of the tumor-bearing mice after the tail vein injection of free Pheo a and PPHE polymeric nanoparticles and (**b**) quantification of the average fluorescence signals in the tumor site. (**c**) Ex vivo fluorescence images of the tumor and normal organs (i.e., liver, lung, spleen, kidney, and heart) excised from the tumor-bearing mice at 24 h post-injection and (**d**) quantification of the average fluorescence signals of the tumor and normal organs. * *p* < 0.05 for comparison between two treatment groups.

## Data Availability

The data presented in this study are available on request from the corresponding author.

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
