# Peer review of "Photodynamic and Cold Atmospheric Plasma Combination Therapy Using Polymeric Nanoparticles for the Synergistic Treatment of Cervical Cancer"

_ijms, 2021, doi:10.3390/ijms22031172_

Round 1
Reviewer 1 Report
The authors present a study on multimodal therapy approach for cervical cancer. The study is of hot topic, is very sound and well written. Nevertheless, I have a problem with many missing controls in the experiments, which is quite severe thing not to include appropriate controls, which, unfortunately, significantly reduces the overall quality of otherwise very interesting and very good study. Also, and very importantly, discussion is fully missing
Things to improve:
line 54 - PDT is definitely not selective type of treatment (in general, there are special cases - like using targeting moieties, but in general it is not). Moreover, yourself contradict the statement in the line 59.
line 55 - yes, it does also damage the adjacent tissues
line 62 - examples of specific targeting has been shown, e.g. to the androgen receptor, the following citation is missing and should be added:
Steroids 2015 May;97:62-6. doi: 10.1016/j.steroids.2014.10.002. Epub 2014 Nov 1. Synthesis and biological evaluation of nandrolone-bodipy conjugates
Figure 1 - the caption could be much more explanatory/descriptive
Figure 3 - the caption - the length of treatment and the irradiation condition (source, W, time) are fully missing and must be included
Figure 5 - fluorescence intensity - untreated control missing, positive control (hydrogen peroxide) missing, control of Pheo nonphotoactivated missing, etc., all needed controls (there are many more) are fully missing, they must be a part of the experiment
Line 248 - what does CAT mean?
Line 316 - you used MTT for evaluation of viability of 3D cell model after treatment, how does the MTT gets inside of the 3D model, it is known that the drug (as well as nutrients) penetration is very poor. Also, some sites back, you claim that there is mitochondrial damage. How can you use MTT for viability determination when mitochondria are damaged?
Methods - cell culture - why did you used antibiotics??
chapter 3.6 - line 426 - the cells were rinsed and then what? they were irradiated in dry state or how??
details for the MTT assay are missing
DISCUSSION is fully missing in whole article, but it is almost the most important part of the manuscript, it must be added
Chapter 3.9. DCF-DA assay - details fully missing, controls missing
chapter 3.10 - controls missing - cisplatin as positive one, Pheo in dark, CAP alone, vehicle control, untreated cells as control, PPHE in dark, etc.
Author Response
The authors present a study on multimodal therapy approach for cervical cancer. The study is of hot topic, is very sound and well written. Nevertheless, I have a problem with many missing controls in the experiments, which is quite severe thing not to include appropriate controls, which, unfortunately, significantly reduces the overall quality of otherwise very interesting and very good study. Also, and very importantly, discussion is fully missing.
Things to improve:
Point 1: line 54 - PDT is definitely not selective type of treatment (in general, there are special cases - like using targeting moieties, but in general it is not). Moreover, yourself contradict the statement in the line 59.
- Response 1: Thank you for pointing out our mistake. The main advantages of the PDT treatment are the noninvasiveness and minimal side effects such as the minimization of damage to adjacent normal tissues. This treatment modality usually exhibits limited tumor cell selectivity. Thus we changed “selectivity of cell destruction” to “noninvasiveness”.
Point 2: line 55 - yes, it does also damage the adjacent tissues
- Response 2: As commented by the reviewer, the PDT treatment also reveals side effects such as damage to the adjacent normal tissues, but these side effects are not serious compared to the common cancer therapies such chemotherapy and radiotherapy.
Point 3: line 62 - examples of specific targeting has been shown, e.g. to the androgen receptor, the following citation is missing and should be added: Steroids 2015 May;97:62-6. doi: 10.1016/j.steroids.2014.10.002. Epub 2014 Nov 1. Synthesis and biological evaluation of nandrolone-bodipy conjugates
- Response 3: According to the comment of the reviewer, we added new reference in end of line 62. Thank you for kind comment.
Point 4: Figure 1 - the caption could be much more explanatory/descriptive
- Response 4: As pointed out by the reviewer, we added some sentences to provide more explanatory and descriptive caption of Figure 1.
Point 5: Figure 3 - the caption - the length of treatment and the irradiation condition (source, W, time) are fully missing and must be included
- Response 5: As pointed out by the reviewer, we added the irradiation conditions including laser source and treatment time in the caption of Figure 3.
Point 6: Figure 5 - fluorescence intensity - untreated control missing, positive control (hydrogen peroxide) missing, control of Pheo a nonphotoactivated missing, etc., all needed controls (there are many more) are fully missing, they must be a part of the experiment
- Response 6: As commented by the reviewer, we added the fluorescence intensity of untreated control, unirradiated Pheo a, and unirradiated PPHE in Figure 5b (changed to Figure 4b) as various controls to compare with the results from the PDT treatment and PDT/CAP combinatory treatment using free Pheo a or PPHE polymeric nanoparticles.
Point 7: Line 248 - what does CAT mean?
- Response 7: Thank you for pointing out our mistake. CAT was a mistyping and thus we changed “CAT” to “CAP”.
Point 8: Line 316 - you used MTT for evaluation of viability of 3D cell model after treatment, how does the MTT gets inside of the 3D model, it is known that the drug (as well as nutrients) penetration is very poor. Also, some sites back, you claim that there is mitochondrial damage. How can you use MTT for viability determination when mitochondria are damaged?
- Response 8-1: As commented by the reviewer, the intracellular uptake of MTT in the inside of the 3D model is probably lower compared with the 2D model because of poor drug penetration. Therefore, we checked the accuracy of the MTT assay in the 3D model by comparing with the cell counting method using various type of cell lines (cancer cells and normal cells). Resultingly, the significant difference between the MTT assay and the cell counting method was not observed in all type of cells tested. In particular, the cells treated with drug and laser/CAP exhibited lower viability and thus the intracellular uptake of the MTT in the inside of the 3D model was presumably enhanced. Therefore, we adopted the MTT assay for measuring the cell viability.
- Response 8-2: PDT treatment mainly induces apoptotic cell death due to the produced ROS after irradiation. These ROS can induce the oxidation of a large range of biomolecules in cells, including proteins, DNA, and lipids. In other word, the rapid ROS generation in cancer cells causes mitochondrial damage, release of proapoptotic proteins into cytosol, and DNA fragmentation, resulting in apoptotic cell death. Thus, mitochondrial damage induced by the produced ROS leads to the cancer cell death. We did not measure the dead cell due to mitochondrial damage but measured live cells via the MTT assay.
Point 9: Methods - cell culture - why did you used antibiotics??
- Response 9: The antibiotics, penicillin and streptomycin, were used to prevent bacterial contamination of cell culture due to their effective combined action against gram-positive and gram-negative bacteria. These antibiotics are usually used in cell culture.
Point 10: chapter 3.6 - line 426 - the cells were rinsed and then what? they were irradiated in dry state or how??
- Response 10: The cells were treated with drug (Pheo a or PPHE) for 2 h and rinsed twice with DPBS. Subsequently, fresh culture medium was added to each well and then the cells were treated with a laser and CAP. Thus, the cell were irradiated in wet state in most of the cell culture experiments such as phototoxicity (MTT), LIVE/DEAD, apoptosis and necrosis analysis, and western blot assays. We added one sentence to explain detail of the cell state during the PDT and CAP treatment in pages 13-16.
Point 11: details for the MTT assay are missing
- Response 11: As pointed out by the reviewer, we added some sentences to explain the detail for the MTT assay in pages 13 and 15, “3.6. In vitro Phototoxicity Assays” and “3.12. 3D Cancer Cell Culture Model” sections. Thank you for your kind advice.
Point 12: DISCUSSION is fully missing in whole article, but it is almost the most important part of the manuscript, it must be added
- Response 12: We wrote “Discussion” part together “Results” part. Thus the discussion part was inserted into “2. Results and Discussion” section. As pointed out by the reviewer, the discussion part is very important part in the manuscript and we added this part into “Results and Discussion” section.
Point 13: Chapter 3.9. DCF-DA assay - details fully missing, controls missing
- Response 13: As pointed out by the reviewer, we added some sentences to explain the detail for the DCF-DA assay in page 14, “3.9. Evaluation of the Intracellular ROS Generation” section. In addition, we added one sentence to explain the used control, and the fluorescence intensity for untreated control, unirradiated Pheo a, and unirradiated PPHE was inserted in Figure 5 (changed to Figure 4).
Point 14: chapter 3.10 - controls missing - cisplatin as positive one, Pheo in dark, CAP alone, vehicle control, untreated cells as control, PPHE in dark
- Response 14: According to the comment of the reviewer, we added one sentence in page 15 to explain the control used. In general, untreated cells (no drug and no light) were used as control (vehicle control) to analyze the apoptosis and necrosis of targeted cells in studies for the PDT and PTT treatments of cancer cells (ref. [16], [24], and [Acta Biomaterialia 2018, 82, 171-183]). Therefore, we also used untreated cells as control (vehicle control) as shown in Figure 6 for apoptosis and necrosis analysis of CaSki cells treated with the PDT/CAP combinatory therapy.
Thank you for reviewing our paper.

Reviewer 2 Report
- In this research, although a lot of experiments had been performed by author to prove the research design have great potency in chemoresistance, to confirm the effectiveness of your design you should study Histopathological characterization and estimate release property of nanocarrier.
- The author should mention the cell line used in creating animal model to estimate biodistribution of nanocarrier.
- To determine the physicochemical properties of nanocarrier, TEM imaging is crucial parameter. In this research this TEM imaging were not revealed.
- Sequences of revealed materials and methods, results and discussion were not link each other, for example after for formulation had been done, you should reveal the physicochemical properties of formulation and then, in vitro cellular studies followed by in vivo
Author Response
Point 1: In this research, although a lot of experiments had been performed by author to prove the research design have great potency in chemoresistance, to confirm the effectiveness of your design you should study Histopathological characterization and estimate release property of nanocarrier.
- Response 1-1: Thank you for good advice. We also want to try the in vivo animal testing including histopathological assay to confirm therapeutic efficacy of the PPHE polymeric nanoparticles. Unfortunately, animal testing should be carried out only at good laboratory practice (GLP) center in our university but not at our laboratory. However, we could not rent empty space of GLP center and research equipment. We can use GLP center from seven months later.
- Response 1-2: As commented by the reviewer, we performed the drug (Pheo a) release test from the PPHE polymeric nanoparticles in different pH conditions (pH 7.4 and 4.5). A dialysis method was used to evaluate the release behavior of Pheo a from the PPHE polymeric nanoparticles in a thermostatic shaking incubator (200 rpm, 37 °C). As indicated in Figure S3, The PPHE polymeric nanoparticles exhibited an initial burst release of Pheo a, followed by a prolonged gradual increase of release. Pheo a was released faster from the PPHE polymeric nanoparticles at pH 4.5 than at pH 7.4 in the DPBS solution because of pH-dependent cleavage of amide bonds. Thus we added some sentences to explain the result of cumulative Pheo a release in page 4.
Point 2: The author should mention the cell line used in creating animal model to estimate biodistribution of nanocarrier.
- Response 2: We already mentioned the cell line (CaSki cells) used in animal test in page 18, “3.12. Biodistribution and Imaging Assays” section.
Point 3: To determine the physicochemical properties of nanocarrier, TEM imaging is crucial parameter. In this research this TEM imaging were not revealed.
- Response 3: As commented by the reviewer, we carried out TEM observation and changed SEM image to TEM image of the PPHE polymeric nanoparticles in Figure 2.
Point 4: Sequences of revealed materials and methods, results and discussion were not link each other, for example after for formulation had been done, you should reveal the physicochemical properties of formulation and then, in vitro cellular studies followed by in vivo
- Response 4: As pointed out by the reviewer, we changed sequences in “Materials and Methods” and “Results and Discussion” as follows; Preparation and characterization of the PPHE polymeric nanoparticles, in vitro cell studies, and then in vivo animal study.
Thank you for reviewing our paper.

Reviewer 3 Report
In the manuscript entitled “Photodynamic and Cold Atmospheric Plasma Combination Therapy Using Polymeric Nanoparticles for the Synergistic Treatment of Cervical Cancer” the authors report on the application of a photodynamic therapy(PDT)/cold atmospheric plasma (CAP) combinatory treatment of human papillomavirus (HPV)-positive cervical cancer by using innovative PPHE polymeric nanoparticles.
The rationale of this scientific work is innovative, documented by appropriate literature and based on the very recent achievements obtained in the tumor-targeted therapeutic treatment of the cervical cancer. Moreover, data obtained by in vitro experiments carried out both on 2D and 3D CaSki cell cultures are well presented and discussed.
In my opinion, this manuscript is an example of excellent scientific research, so I recommend its publication in the International Journal of Molecular Science as is.
Author Response
- We carefully checked our manuscript several times to find mistyping and corrected. Thank you for reviewing our paper and high evaluation of our work.

Round 2
Reviewer 1 Report
The authors have answered all questions and also improved the manuscript according tho the reviewer advice. The article can be now accepted.
Reviewer 2 Report
For further study, to modify research design and in vivo study
To confirm drug release and toxicity of nanocarrier.